# Molecular Characterization of Outer Capsid Proteins VP5 and VP7 of Grass Carp Reovirus

**DOI:** 10.3390/v14051032

**Published:** 2022-05-12

**Authors:** Fuxian Zhang, Diangang Sun, Qin Fang

**Affiliations:** 1College of Animal Science, Yangtze University, Jingzhou 430023, China; zhangfuxian99@163.com (F.Z.); sundiangang0330@163.com (D.S.); 2State Key Laboratory of Virology, Wuhan Institute of Virology, Chinese Academy of Sciences, Wuhan 430071, China

**Keywords:** aquareovirus, GCRV, outer capsid protein VP5 and VP7, mass spectrometry and posttranslational modifications, tryptic digestion

## Abstract

Aquareovirus, which is a member of the Reoviridae family, was isolated from aquatic animals. A close molecular evolutionary relationship between aquareoviruses and mammalian orthoreoviruses was revealed. However, the functions of the aquareovirus genome-encoded proteins are poorly understood. We investigated the molecular characteristics of the outer capsid proteins, namely, VP5 and VP7, of grass carp reovirus (GCRV). The peptides VP5 and VP7 were determined using in-gel tryptic digestion and mass spectrometry. Recovered peptides represented 76% and 66% of the full-length VP5 and VP7 sequences, respectively. Significantly, two-lysine acetylation, as well as two-serine and two-threonine phosphorylation modifications, were first revealed in VP5. We found that the initial amino acid in VP5 was Pro43, suggesting that a lower amount of VP5 remained uncleaved in virions at the autocleavage site (Asn42-Pro43). Further biochemical evidence showed that the cleaved VP5N/VP5C conformation was the major constituent of the particles. Moreover, early cleavage fragments of VP7 and enhanced infectivity were detected after limited tryptic digestion of GCRV, indicating that stepwise VP7 cleavage is essential for VP5 conformational rearrangement. Our results provide insights into the roles of posttranslational modifications in VP5 and its association with VP7 in the viral life cycle.

## 1. Introduction

Viruses in the family Reoviridae are non-enveloped particles comprising a segmented double-stranded RNA (dsRNA) genome surrounded by an inner core and an outer capsid protein shell. Core capsid proteins are generally involved in facilitating endogenous transcriptional activity and viral replication [1], whereas outer capsid proteins (OCPs) are critical for nascent particle morphogenesis and initiating infection during cell entry [2,3]. Studies on mammalian orthoreoviruses (MRVs) have revealed that the capsid protein μ1 is N-terminally myristoylated and mediates the penetration of cellular membranes during entry [4]. The σ3 zinc metalloprotein occupies its outermost position by tightly binding μ1 to form a heterohexameric complex on the viral capsid, allowing it to maintain viral particle stability [2,5]. In addition, the μ1 protein was also shown to play a role in transcriptase activation [6]. σ3 is able to bind dsRNA and block the activation of interferon-induced and dsRNA-activated protein kinase R (PKR). Additional studies suggested roles for σ3 in the regulation of viral transcription and translation [7]. Therefore, μ1 and σ3 are multifunctional proteins and their associations play an indispensable role in their functions and viral replication cycle.

Aquareoviruses, which are members of the Spinareovirinae group in the Reoviridae family, mainly infect aquatic animals, including fish and shellfish, and pose a serious threat to the aquaculture industry worldwide [8]. Many aquareoviruses were isolated from fresh and salt waters [9,10,11,12]. Aquareoviruses can produce large syncytia as a typical cytopathic effect (CPE) in permissive cell cultures. Generally, aquareovirus infection causes low pathogenicity in farming aquatic cultures, which is often detected during routine examinations. However, grass carp reovirus (GCRV), which causes severe hemorrhagic disease in grass carp fingerlings and yearlings, is considered one of the most important pathogens in fish breeding [13,14]. Many new reoviruses were isolated from diseased and healthy grass carp [15,16]. Three types of GCRVs (GCRV-I, -II, and -III) were genetically classified based on sequence alignments of the S8 genome segment that encodes the core clamp VP6 [17,18,19,20]. Except for GCRV104, which is the only member in GCRV-III, notable differences are observed between GCRV-I (type strain, GCRV873) and GCRV-II (type strain, GCRVHZ08) isolates on pathogenic symptoms and predicted genome-encoded protein characteristics. GCRV-I and -II can be divided into fusogenic and non-fusogenic strains based on the CPE in respective cell cultures. In this regard, GCRV-I is recognized as a fusogenic aquareovirus, which represents the majority of aquareoviruses. GCRV-II is supposedly a non-fusogenic reovirus because it harbors a predicted MRV-σ1 cognate cell attachment protein on the particle surface instead of a fusion-associated small transmembrane protein encoded by S7. Despite host and genus differences between aquareoviruses and MRVs, they are highly related and share nine genome-encoded homologous proteins [21].

Similar to MRV, the outer capsid of GCRV is composed of VP5 trimers, which are clamped by three copies of VP7, forming a VP5–VP7 heterohexameric complex. As the homologous protein of μ1, the overall structure of VP5 in aquareoviruses appears to be very similar to that of μ1, suggesting that the OCP VP5 in aquareoviruses may have functions similar to those of μ1 [22,23,24]. Indeed, the VP5 atomic model, resolved using cryo-electron microscopy (cryo-EM) and three-dimensional (3D) reconstruction, was very similar to the μ1 protein crystal structure [25]. Furthermore, the myristoyl groups covalently linked to the VP5 N-terminus were resolved using high-resolution 3D imaging of GCRV, indicating that aquareoviruses use the same membrane penetration mechanism as MRV and other non-enveloped viruses for cell entry [25,26]. Other studies have revealed that the capsid protein VP5 shares marked structural similarities with members of the Orthoreovirus genus, including avian reoviruses, baboon reovirus, and MRVs, suggesting similar functions between VP5 and its homologous μ1/μB [27]. In addition to VP5, overall protein structural similarities were found in the outermost protein σ3 of the MRV and aquareovirus VP7, which contradicts the low sequence identity (~12%) obtained in their sequence alignment [22]. Bioinformatics analysis showed that all VP7 proteins in the aquareovirus and the corresponding homolog σ3 in the orthoreovirus species contain the CCHC zinc-binding motif at the N-terminal region, suggesting that both proteins have partially similar functions [1]. In striped bass reovirus (SBRV) and GCRV, in vitro protease cleavage of the outer shell protein VP7 can lead to enhanced infectivity [28,29]. Therefore enhancement of infectivity is correlated with the digestion of the outer capsid protein VP7 in viral particles. Considerable progress has been made to reveal the structural information of aquareovirus functional proteins during replication and cell entry [25]. Nonetheless, a comprehensive understanding of how aquareoviruses initiate infection and detailed events in virus-infected cells leading to OCP (VP5/μ1 and or VP7/σ3) assembly onto nascent cores remains to be fully elucidated.

GCRV873 (prototype GCRV strain, named GCRV throughout this paper) has been extensively used to understand aquareovirus molecular and structural biology. Seven structural (VP1–VP7) and six nonstructural proteins (NS12, NS16, NS26, NS31, NS38, and NS80) encoded by the GCRV 11-segment dsRNA genome were identified [9,30,31,32,33]. In particular, seven GCRV structural proteins and their respective localization in viral particles were resolved using cryo-EM and 3D reconstruction with a high resolution [22,23,25,34]. Myristoylated modifications embedded in the hydrophobic pockets of the VP5 N-terminus indicate that posttranslational modifications (PTM) are critical for VP5 conformational transformation and membrane penetration [26]. In this study, the molecular characteristics of VP5 and VP7 were analyzed using liquid chromatography-tandem mass spectrometry (LC-MS/MS), and the related biochemical properties were investigated to understand the biological function of VP5 and its coordinated role with VP7.

## 2. Materials and Methods

### 2.1. Cells, Virus, and Antibodies

Ctenopharyngodon idellus kidney (CIK) cells from the China Center for Type Culture Collection (CCTCC, 4201FIS-CCTCC00086) were grown at 28 °C in Eagle’s minimum essential medium (MEM, Invitrogen, USA), and Spodoptera frugiperda (Sf9) cells (from CCTCC, 4201INS-CCTCC00008) were cultured in Grace medium at 28 °C (Gibco BRL, Rockville, USA). Both cell culture media were supplemented with 10% fetal bovine serum (Gibco BRL, Rockville, MD, USA). GCRV873 (prototype GCRV strain) was grown in monolayers of CIK cells and used for infection and propagation, as previously described [23]. Sf9 cells were used to propagate the recombinant baculoviruses.

Mouse and rabbit polyclonal antibodies against the OCPs VP5 and VP7 were prepared as reported previously [35]. Alkaline phosphatase-coupled goat anti-mouse IgG and goat anti-rabbit IgG were purchased from Sigma-Aldrich ( Burlington, MA, USA). Alexa Fluor^®^ 568 donkey anti-mouse IgG (H+L) (red) and Alexa Fluor^®^ 488 donkey anti-rabbit IgG (H+L) (green) were purchased from Invitrogen Co. (Carlsbad, CA, USA).

### 2.2. Virus Particle Purification

GCRV particles from the infected culture supernatant were purified by combining different centrifugations and cesium chloride (CsCl) density or sucrose gradient centrifugation methods established in our laboratory [29]. GCRV particles were extracted from the collected culture supernatants by differential centrifugation. Cells were centrifuged at 5000× g for 30 min at 4 °C to remove cell pellets, followed by ultracentrifugation at 80,000 g in an SW28 rotor (Beckman, USA) for 2 h at 4 °C to harvest the virus. For further purification, the viral pellet was resuspended and layered on a ~12 mL (SW41 Ti) CsCl gradients (ρ, 1.20–1.45 g/cm^3^) tube. This tube was centrifuged at 4 °C in an SW41 Ti rotor (Beckman, USA) at 230,000 g for 3.5 h. After centrifugation, separated viral bands were formed. After carefully removing the separated virus bands, each viral component was further resolved using 1× phosphate-buffered saline (PBS) in SW55 tubes and centrifuged at 210,000 g for 30 min to remove residual sucrose or salts. The pelleted virions were resuspended in PBS to a concentration of approximately 1.5 mg/mL for either transmission electron microscopy (TEM) or cryo-EM sample preparation. The final concentrations of the purified fresh virion suspensions were determined by measuring the optical density using a spectrophotometer (Biotek, Winooski, VT, USA) at 260 nm [29,36]. All the purified samples were stored at either 4 °C, −20 °C, or −80 °C for further usage.

### 2.3. Generation of Recombinant Baculoviruses to Express VP7 and VP5/VP5^N42A^

The recombinant baculoviruses containing the VP7 gene (S10), as well as the wild-type (S6) and mutant VP5^N42A^ gene from GCRV873, were described previously [37]. To generate a recombinant baculovirus expressing VP5 or VP7 individually or together, the S6 and S10 GCRV genes were cloned into the pFastbacI or pFastbac-dual vector (Invitrogen, USA), respectively, as previously reported [37]. The primer pairs were designed based on GenBank sequences (AF403392 and AF403396) and are available upon request. The positive recombinant plasmid was transformed into DH10Bac™-competent cells (Invitrogen, Carlsbad, CA, USA) according to the manufacturer’s instructions. The isolated recombinant bacmids (pFbDGCRV-VP5/VP7, pFbIGCRV-VP5, pFbIGCRV-VP7, and pFbDGCRV-VP5^N42A^/VP7) were verified via polymerase chain reaction (PCR) with either M13 primers or gene-specific primers, as described previously [38].

### 2.4. In Vitro Expression of VP5 and VP7 Proteins

Recombinant bacmids pre-mixed with the cationic liquid Cellfectin reagent were transfected into monolayers of Sf9 cells according to the manufacturer’s instructions (Invitrogen, Carlsbad, USA). Three to five days post-transfection, the cell suspension was harvested following the previously described protocol [39]. The harvested supernatant, named Passage1 (P1), was used for further inoculation of Sf9 cells until a stable recombinant baculovirus was generated. To express VP5 and VP7 alone or co-express wild-type VP5 or mutant VP5^N42A^ with VP7, Sf9 cells were infected at a multiplicity of infection (MOI) of 5 plaque-forming units (PFU)/cell with fourth passage recombinant baculovirus stocks. The expression or co-expression cell lysates were harvested 72 h post-infection (p.i.), and cytoplasmic extracts were prepared, as described previously [37,39]. To express VP7 with VP5 or/VP5^N42A^ and for their large-scale co-expression, Sf9 cells were inoculated with P4 or P5 virus stocks. To prevent expressed protein degradation, the protease inhibitors aprotinin and leupeptin (0.5 mg/mL each) were added at 1 or 2 days p.i., and the infected cells were harvested at 3 days p.i. and stored at −80 °C for further use.

### 2.5. SDS-PAGE and Immunoblotting

The samples used for protein component analysis were subjected to sodium dodecyl sulfate-polyacrylamide gel electrophoresis (SDS-PAGE), as described previously [29]. The separated protein bands were visualized via staining with Coomassie Brilliant Blue R-250 (Sigma, Ronkonkoma, NY, USA). Ten or 12% acrylamide gels were used for SDS-PAGE, unless otherwise specified.

Immunoblotting (IB) was performed as previously described [30]. Briefly, all samples were first subjected to PAGE and then transferred to a polyvinylidene fluoride (PVDF) membrane. After blocking with 3% bovine serum albumin (BSA) in PBS with 0.05% Tween 20 (PBST), the membrane was incubated with rabbit/mouse anti-VP5 and VP7 polyclonal antibodies and then subjected to the secondary antibody conjugated with alkaline phosphatase (AP). Reactive bands were developed using AP substrate solution (NBT/BCIP).

### 2.6. Immunofluorescence and Fluorescent Focus Assay

Immunofluorescence (IF) assays were performed as previously described [32,40]. CIK cells were seeded and cultured in glass-bottom Petri dishes overnight at 28 °C, then infected with GCRV viral particles at an MOI of 5. After completing the essential procedure, the samples were incubated with mouse or rabbit polyclonal antibodies against VP5. Alexa-488-labeled donkey anti-mouse IgG (H+L) was used as the secondary antibody. A fluorescent focus assay was performed to assess viral replication. Fluorescent foci of infection were observed, and viral titers were calculated as fluorescent focus units (FFU) per mL by counting using a fluorescence microscope. Infected cells were quantified by counting random fields in equivalently confluent monolayers for at least five fields of view in triplicate wells [40].

### 2.7. Trypsin Digestion, LC-MS/MS Analysis, and Data Analyses

VP5 and VP7, which were prepared using in-gel digestion and extraction of target peptides, were performed with trypsin as described elsewhere [41]. The protein bands excised from SDS-PAGE were cut and de-stained via repeated alternating washes with 100 mM NH_4_HCO_3_, including a 1:1 (v/v) mixture of CH_3_CN and 100 mM NH_4_HCO_3_ solutions. Briefly, the protein samples to be tested were diluted after adding 100 mM NH_4_HCO_3_ to a urea concentration of less than 2 M. Subsequently, two rounds of trypsin (Promega, Madison, CT, USA) digestion were conducted. For the initial cleavage, trypsin was added at a 1:50 trypsin-to-protein mass ratio for the first digestion at 37 °C overnight, and then a 1:100 trypsin-to-protein mass ratio for the second two-hour digestion was performed. The fragmented VP5 and VP7 peptides were desalted, vacuum-dried after trypsin digestion, and reconstituted in 0.1 M NaAc (pH = 5.99) as described previously [41]. For the LC-MS/MS analysis, peptide samples were dissolved in 0.1% formic acid and directly loaded onto a reversed-phase pre-column (Acclaim PepMap 100, Thermo Fisher Scientific, Waltham, MA, USA). A reversed-phase analytical column (Acclaim PepMap RSLC, Thermo Fisher Scientific, Waltham, MA, USA) was used for peptide separation as previously reported [42]. The resulting peptides were analyzed and intact peptides were detected using Orbitrap (Thermo Fisher Scientific, Waltham, MA, USA). For the LC-MS/MS assays, the ion fragments detected in the Orbitrap were set up at a resolution of 17,500. For the MS scans, the m/z scan range was set as 350 to 1800. Peptide sequences were determined by matching the acquired fragmentation patterns with protein databases using the Sequest software (Thermo Finnigan, Mundelein, IL, USA). Tandem mass spectra were searched against the open reading frame database of GCRV for viral protein identification. The mass error was set to 5 ppm for the main search and 0.02 Da for fragment ions. The other parameters in MaxQuant were set as previously described [42]. The protein acetylation site and phosphorylation site identification were processed using MaxQuant with the integrated Andromeda search engine (v. 1.4.1.2). The lysine acetylation site and phosphorylation site localization probabilities were set at >0.75. All experiments were repeated three times.

### 2.8. Limited Early Digestion of Virions with Trypsin

The limited early digestion of the virions with trypsin was modified from our previously established method [29]. Purified GCRV virions at approximately 1.5 mg/mL in a virion suspension buffer (150 mM NaCl, 10 mM MgCl_2_, 10 mM Tris, pH 7.5) were digested with trypsin at a final concentration of ~100 μg/mL (total volume: 20 μL) at 28 °C to generate trypsin-treated GCRV particles (TTP) by treating for 15 min (15m-TTP) then 30 min (30m-TTP). Meanwhile, mock-treated native particles (NP) served as a control, and the 15m-TTP, 30m-TTP, and NP preparations were directly applied to conduct electron microscopy, SDS-PAGE analysis, and the inoculation of CIK cells for infectious assays without further purification. For the biochemical analysis of 15m-TTP and 30m-TTP, purified GCRV digested with chymotrypsin (Bovine pancreas, Sigma) were used as parallel controls, which were incubated at 28 °C for 30 min to generate chymotrypsin (final concentration: ~100 μg/mL)-treated particles (30m-ChTP). All incubations were terminated by placing the reaction mixtures on ice.

### 2.9. Transmission Electron Microscopy and Cryo-EM Image Preparation

To examine the morphology of purified viral particles, the purified fresh virion preparations were loaded onto carbon-coated copper grids (200-mesh) for a 5 min incubation at room temperature and negatively stained with 3% (*w*/*v*) phosphotungstic acid (pH 6.8) for 1 min. After drying overnight, all the sample grids were examined using TEM (Hitachi, Tokyo, Japan, H-7000FA). For cryo-EM, the prepared viral specimen (~3.0 μL/each) was applied to one side of a holey carbon grid and then quickly plunged into a bath of liquid ethane cooled by liquid nitrogen using an FEI Vitrobot (Thermo Fisher Scientific, MA, USA). The frozen-hydrated samples were transferred to a precooled GATAN cryo-holder and observed using a JEOL 1200 transmission electron microscope operated at 200 kV and maintained at liquid nitrogen temperature.

## 3. Results

### 3.1. GCRV Purification and Structural Protein Analysis

Previous studies showed that GCRV is composed of seven structural proteins, including five core proteins (VP1-VP4 and VP6) and two OCPs (VP5 and VP7). To define structural protein properties, highly purified virions from infected cell cultures are required. Using the established GCRV purification protocol, CsCl density gradient centrifugation was performed to obtain purified viral particles from a concentrated virus-cell mixture. Two obvious homogeneous bands with an opalescent appearance (upper and lower bands) were formed in the CsCl gradient centrifugation tube (Figure 1A). The two bands corresponded to approximately 1.30 and 1.38 g/cm^3^ in buoyant density, respectively, indicating that the viral components were successfully separated after CsCl gradient centrifugation. To define the particle forms and protein components of purified GCRV, negatively stained TEM and SDS-PAGE analyses were performed. As shown in Figure 1B, the intact virions obtained from the lower band exhibited an overall even morphology with a very clean background. The empty particles or top components from the top band appeared similar in shape to intact virions but presented a central cavity, indicating the absence of genomic dsRNA in the interior. Protein gel analysis identified seven structural protein components (VP1 to VP7) in both intact and empty particles (Figure 1C). In comparison with intact particles from the lower band, the outermost protein, namely, VP7, was somewhat degraded in the empty particles located in the upper band, which was consistent with the observation of a few protein subunits scattered in the TEM field with top components (Figure 1B upper panel). Collectively, the protein components of both the intact and empty virions were comparable. The highly purified VP5 and VP7 proteins of intact virions were suitable for MS assays.

### 3.2. MS Assays of GCRV OCP VP5

The genome sequence indicated that VP5 was composed of 648 amino acids (aa) encoded by the S6 segment (2039 nucleotides (nt)) [43]. LC-MS/MS assays were performed to understand the properties and function of VP5. Using in-gel tryptic digestion, too many tryptic fragments of VP5 were obtained, and a large number of repeatedly cleaved peptides were identified (data not shown). As shown in Appendix A, the longest tryptic fragment in VP5 was 44 residues, from 473 to 516 (ETT…LNK), and the shortest fragment was 7 aa. Amongst the 38 ion fragments identified, one was not cut at the specific tryptic cleavage site, while the remaining 37 listed peptide fragments were all cleaved at the specific cleavage site (after R and/or K), which is identical to trypsin digestion. Analyses of these VP5 tryptic fragments showed (Figure 2A) that they accounted for 492 of the 648 predicted aa identified in VP5 (76% coverage). A selected identified peptide spectrum map from residues 417 to 428 (FNMLHLQATFER) with an observed m/z value of 513.6077 (ion score: 43.337) is shown in Figure 2B. Of the six unidentified peptide fragments, two shorter unidentified regions (from residues 413–416 and 569–570) were cleaved by trypsin to generate the tetrapeptides GQPR and dipeptides AK, respectively, which were too small to be identified using MS. The other longest unidentified region constituting 66 residues was from residues 347 to 403 (VID……GAS). Careful examination revealed no trypsin cleavage site in the region, thus it was not identified. Recovered peptides represented 76% of the VP5 sequence excluding N-terminal residues 1-42 (MGN……VLN), which contained one trypsin cleavage site K37, even though our MS was repeated a few times. Indeed, only Pro43 after the autocleavage junction was the initial residue identified in our MS assays, which was consistent with the observation that the Asn42-Pro43 bond in VP5 was cleaved [25,26]. This finding provided additional evidence that full-length VP5 was cleaved into VP5N and VP5C. The fact that the N-terminus 1–42 residues were not identified by LC-MS/MS suggested that full-length VP5 was present at a lower level in particles than cleaved VP5N and VP5C.

The myristoyl group linked to the N-terminal aa of the VP5 protein was observed in an atomic model using cryo-EM of GCRV [25,26]. Considering that protein acetylation and phosphorylation are ubiquitous among diverse species, to determine whether there were other PTMs in VP5, phosphorylated and acetylated modifications were analyzed. As shown in Table 1, six sites of the VP5 PTMs were identified in the five cleavage peptide fragments. Among these modifications, four phosphorylated and two Lys-acetylated modifications were identified (Figure 3). Compared with the resolved VP5 structure, the identified lysine acetylation sites (Kac219 and Kac623) were located at the base and link domains, which suggested that the acetylated modifications may play roles in regulating nascent particle assembly. The two phosphorylated serine residues at Ser283 and Ser556 were located in the link domain. Another two phosphorylated modifications, namely, Thr165 and Thr171, were located in the base domain (Figure 3A). Of the six identified PTMs in VP5, the notable LC-MS/MS spectrum of a tryptic peptide ion Ser556 phosphorylated peptide detected as a doubly charged precursor ion with an m/z value of 1103.0388 (ion score: 106.70) is shown in Figure 3B. Given that Ser556 phosphorylation is within the 551–560 loop region containing the δ-φ protease cleavage site [26], it is likely that Ser556 phosphorylation plays a role in regulating the activity of the myristoyl insertion finger by releasing the φ fragment and causing VP5 conformational rearrangement during cell entry. Due to the VP5 localization in particles, it has a close interaction with complex VP7, also interacting with inner core clamp VP6 and turret protein VP1 to maintain particle strength and stability [22,34]. The identified modifications of Kac219, Kac623, and four phosphorylated modifications (Ser283ph, Ser556ph, Thr165ph, and Thr171ph) may play critical roles in modulating particle assembly and disassembly and are beneficial for VP5 conformational changes during cell entry.

### 3.3. MS Assays of the GCRV Outermost Capsid Protein VP7

The smallest structural protein in GCRV is VP7, which is encoded by the S10 genomic segment (909 nt) and comprises 276 aa [43]. The complete atomic model of VP7 in native particles has not yet been resolved due to its outermost localization to protect the penetration protein VP5 and high sensitivity to protease activity. To understand the structure and role of VP7 disassembly during cell entry, VP7 resolved using SDS-PAGE was prepared for in-gel trypsin treatment and analyzed using LC-MS/MS. Using tryptic digestion, 66% MS coverage was obtained, which accounted for 182 of the 276 predicted aa in VP7. The longest identified peptide was 34 aa and the shortest was 8 aa. Analyses of these 12 identified fragments showed that they were all cleaved after lysine and arginine residues. As shown in Appendix A and Figure 4A, the remaining VP7 residue sequences were recovered, except for the N-terminus residues 1–27, the C-terminus residues 235–276, and the other two short peptide fragments in the middle. Two continuous peptide fragment spectra, one from 146 to 166 (*m*/*z*: 595.5674, score: 63.869) and the other from 167 to 184 (*m*/*z*: 547.5541, score: 75.548), are shown in Figure 4B,C. Although the N-terminal aa residues were not identified in this MS assay, VP7 residues (3–88) containing the zinc motif were resolved using cryo-EM of GCRV [25]. These results provided a basis for further investigating VP7 natural sensitivities to protease treatment at different time points.

### 3.4. Autocleavage of VP5 between Residues Asn42 and Pro43 Detected in Native Particles and Lysates of Infected Cells

Considering that only cleaved VP5C fragments were identified using the MS assay, no N-terminal residues (1–42) were identified with few assay repeats. To confirm whether the cleaved VP5N and VP5C conformation is the main architecture in the particle, we excluded the possibility that the cleavage was an experimental artifact due to the reagents used for gradient centrifugation during the particle purification procedure. For this purpose, an additional sucrose gradient centrifugation was performed for comparison with the CsCl gradient centrifugation of purified virions, which were then analyzed using SDS-PAGE. As shown in Figure 5A,B, the protein bands of VP5C, separated from both the CsCl and sucrose gradient centrifugation, appeared more intense than those of full-length VP5, suggesting that the CsCl used for gradient centrifugation was not sufficient to cause VP5 structural conformation changes in the NP of GCRV. To further verify that VP5N and VP5C were generated via autocleavage of VP5 at the Asn42-Pro43 bond, co-expressed VP7 and wild-type VP5 or mutant VP5^N42A^ were analyzed in insect Sf9 cells, as reported previously [37]. Purified and unpurified insect cell lysates containing wild-type VP7 co-expressed with either WT VP5 (vAcGCRV-VP5/VP7) or VP5^N42A^ (vAcGCRV-VP5^N42A^/VP7) were analyzed using IB. As shown in Figure 5C,D, the band of full-length VP5, except for mutant VP5^N42A^/VP7 co-expression, appeared very weak, whereas the bands of VP5C could be easily detected in infected CIK cells and Sf9 cells co-expressing VP5 and VP7. This suggested that cleaved VP5 (VP5N and VP5C) is a major conformation in the lysate of infected cells. In addition, the cleaved VP5δ fragment was detected using IB with mutant VP5^N42A^/VP7 in the lysate of infected cells. This suggested that complex products with mutant VP5^N42A^/VP7 could be cleaved by exogenous proteases from the C-terminal δ-φ proteolytic cleavage site (loop 511–560), which is similar to WT-VP5/VP7 [26]. Similarly, a protease-sensitive conformer of MRV μ1 was detected in the recoated mutant μ1^N42A^ pr-core [3]. The four VP5 conformations detected in this study are shown in Figure 5E. Taken together, it could be concluded that the cleaved VP5 generated at the autocleavage site was a major component of the viral particles.

### 3.5. Cleavage Fragments of VP7 with Enhanced Infectivity were Detected with Limited Early Tryptic Digestion

The 3D images of GCRV revealed that VP7 is located at the outermost position of the aquareovirus, binding tightly with VP5 to form heterodimers in the particle and protect VP5 [22,23]. A modified proteolytic cleavage protocol was conducted to probe the early-stage proteolytic cleavage status of GCRV. This protocol was based on MS sequences covering 66% of VP7, using in-gel tryptic digestion, combined with our previously established protease treatment protocol [29]. For this purpose, trypsin-treated particles at different early time points of 15 min (15m-TTP) and 30 min (30m-TTP) were obtained from purified GCRV. Subsequently, and differently from the previous protocol, the various specimens were directly applied to a quantifoil grid without conducting any further purification for negatively stained TEM, cryo-EM, and infectious analyses [29,44]. Micrographs of negatively stained and unstained frozen-hydrated mock-treated GCRV NP, as well as 30m-TTP, are shown in Figure 6A. Although the overall morphology at 30m-TTP appeared similar to that of the GCRV NP, the densities in the outer shell surface of 30m-TTP appeared slightly weaker than that of the control NP after careful examination, indicating that a few outer shell protein subunits may have been somewhat removed from the periphery in 30m-TTP. Further biochemical analysis showed (Figure 6B) that one main band (~20 kDa) and two weaker bands (~25 and ~18 kDa) of VP7 in TTPs were detected with 15m-TTP and 30m-TTP. Additionally, the two weak bands in 30m-TTP were obviously reduced compared with those in 15m-TTP (lanes 4 and 6 in Figure 6B), which were thought to be intermediate products of tryptic digestion. A comparison of 15m-TTP and 30m-TTP with 30m-ChTP as a protease treatment control showed that the cleaved VP7 fragment with approximately 25 kDa in 30m-ChTP was predicted to be cleaved around the site of Tyr238. As such, the ~20 kDa main intensive band in 15m-TTP and 30m-TTP was predicted to be cleaved at the site of residue 184 (after residue DK). The two remaining weaker bands were predicted to be cleaved at residues 234 (after AR) and 166 (after FK). In addition to the cleaved VP7, the other structural proteins in 15m-TTP and 30m-TTP remained intact, except for very faint bands detected around ~55 kDa, which are supposedly generated at the VP5C/ δ and φ cleavage site at the C-terminus [29]. Indeed, cleaved VP5C/ δ could also be detected in the two top component samples (lanes 1 and 2 in Figure 6B), while no VP5C/ δ cleavage fragment was observed in the NP with intact VP7 (lanes 3 and 6 in Figure 6B). Together, these results suggested that the early tryptic cleavage of VP7 can cause VP5 conformational changes. Given that 15m-TTP is an intermediate product of tryptic cleavage, further infectivity assays were conducted between 30m-TTP and NP. As shown in Figure 6C, the formation of typical fluorescent foci of infection by 30m-TTP in CIK cells was detected at 6 h p.i., which appeared comparatively enhanced compared with the NP. Further time-phase infection analysis indicated that infectivity increased more than 100-fold with 30m-TTP infection compared with mock-treated NP at the early replication stage (Figure 6D). This suggested that 30m-TTP had a better effect on viral infectivity than the NP. Taken together, these results indicated that stepwise VP7 cleavage can generate infectious subviral particles and is essential for VP5 conformational transformation.

## 4. Discussion

Many aquareoviruses were isolated from freshwater and saline waters in recent years, and some of their genome sequences were characterized [10]. Among the isolated aquareoviruses, GCRV was used as a model to understand viral infection and pathogenesis [9]. Notably, almost all structural proteins contained in GCRV were resolved using cryo-EM, except for the full-length VP7 structure. Although previous 3D images of GCRV revealed that VP5N and VP5C were the result of VP5 autocleavage at the site of Asn42-Pro43, and an atomic model of the VP7 residues 3–88 was built in native degraded particles [25], many comprehensive molecular events involved in viral infection, replication, and nascent particle morphogenesis remain largely unknown. In the current study, MS assays and extended biochemical analyses related to VP5 autocleavage and VP7 protease cleavage during infection were conducted to further understand the nature of VP5 and its coordinated association with VP7 in particle-based properties. The results obtained in this study suggest that the VP5 PTMs of phosphorylation and acetylation, as well as the associations and/or disassociations of VP5 and VP7 might play significant biological roles in particle morphogenesis and support VP5 conformational changes during cell entry.

In contrast to the structural biology methodology, which can resolve macromolecular complex structures using cryo-EM and single-particle reconstruction, MS assays provide a powerful tool for the structural characterization of various functional proteins. Moreover, the high accuracy of MS/MS measurements is particularly useful for the identification of PTMs. Using the LC-MS/MS assay, the OCPs VP5 and VP7 protein characteristics and phosphorylation and acetylation PTMs in the VP5 protein were first revealed. In-gel tryptic digestion of VP5 and VP7 yielded high coverage (76% and 66% of the aa sequences, respectively). This further proved the accuracy of our previously defined characteristics of the OCPs VP5 and VP7 [23,43]. Moreover, all recovered VP5 and VP7 peptides were determined via LC-MS/MS using tryptic digestion from purified GCRV, suggesting that a high mass accuracy was obtained in this study. Particularly, analysis of in-gel digestion of the VP5 tryptic fragment revealed that Pro43 was the initial residue that was detected, while the N-terminal (1–42 aa) VP5 sequences were not covered in this identification, which may suggest that the uncleaved VP5 protein component at the putative cleavage site (Asn42-Pro43) appears in small amounts in mature particles. To identify the uncleaved μ1 protein in MRV, the enriched μ1 band was obtained by varying the experimental conditions for the disruption of virions before running proteins on denaturing gels for protease fragmentation and MS. Recovered peptides represented 44% of the full-length μ1 sequence, which contained the peptide from 35 to 48 (SLSPGMLNPGGVPW), spanning the μ1N/μ1C cleavage junction residues [45]. Collectively, the cleaved VP5N/μ1N and VP5C/μ1C in aquareovirus/MRVs might be native conformations in mature particles after particle assembly, which possess biological functions.

PTMs play significant roles in protein function by modulating protein activity, cellular location, and protein-protein interactions. Among all the identified PTMs, acetylation and phosphorylation emerged as key PTMs that play vital functions in controlling the biological processes of various organisms [41,46]. Viral protein phosphorylation was found to regulate vital processes and functions in viral transcription and replication, RNA binding activity, and virus assembly [46,47]. Four phosphorylated sites (Thr165ph, Thr171ph, Ser283ph, and Ser556ph) were first identified in VP5, which have not been reported in its homolog μ1 or in the cognate protein of reoviruses. Phosphorylated Ser556 was located at the C-terminal proteolytic cleavage site [26], indicating that Ser556 phosphorylation plays an important biological role in VP5. In the *Reoviridae* family, phosphorylation of NS2 is identified in Bluetongue virus, which has been demonstrated to control viral inclusion body formation, consistent with a model in which NS2 provides the matrix for viral assembly [46]. In addition, lysine acetylation was also found to regulate viral protein activity and protein-protein interactions to maintain capsid protein stability. Evidence of lysine acetylation was found in the coat protein of cereal yellow dwarf virus-RPV, which is a non-enveloped plant virus in the *Luteoviridae* family. Two conserved residues (Lys147 and Lys181) were identified within an interfacial region, which is presumed critical for virion assembly and stability [48]. Moreover, lysine acetylation sites in Middle East respiratory coronavirus (MERS-CoV) replicase pp1ab indicate that MERS-CoV might use the host acetylation machinery to regulate its enzyme activity and achieve optimal replication [49]. Two lysine acetylations (Kac219 and Kac623) located in the base and link domains were identified in VP5 for the first time in this study, suggesting that lysine acetylation at specific sites is significant for maintaining protein conformation and enabling biological function. Aquareovirus/reovirus core proteins VP1/λ1 and VP6/σ2 were shown to promote the stability of disassembly intermediates and influence early replication events [22,34,50]. The fact that VP5 interacts with VP1 and VP6, as well as VP7 in particles, suggests that VP5 phosphorylation and lysine acetylation at specific sites might play additional roles in regulating nascent particle assembly during viral replication.

The autocleavage of OCPs commonly occurs in several non-enveloped animal viruses. Studies on animal viruses indicate that outer shell protein autocleavage occurs as a maturation step during the assembly process and is required for generating stable, infectious virions [4]. However, a few other studies suggested that viral capsid protein autocleavage occurs at least partially in concert with particle disassembly, which might be one of a series of structural changes in the outer capsid preceding membrane permeabilization in vitro and membrane penetration during cell entry [26,45]. To demonstrate that VP5N and VP5C are major conformations in mature particles, the in vitro co-expression of VP5 and VP7 was analyzed considering that VP5 N-terminal residues (1–42) were not recovered in our MS analysis. Almost all WT-VP5 showed cleaved VP5N and VP5C conformations, whereas mutant VP5^N42A^ was uncleaved, indicating that the cleaved VP5N and VP5C conformations were major components of viral particles and lysates in infected cells. These results clearly demonstrated that aquareovirus VP5 autocleavage was required not only for nascent particle maturation but also for the initiation of efficient infection because there was no absolute dormant status of viruses in infected live host cells. Taken together, our results suggested that autocleavage of aquareovirus VP5 may be linked to particle assembly maturation in the viral replication cycle, even though it also affected disassembly and entry.

The GCRV outer capsid is composed of 600 copies each of VP5 and VP7. Similar to μ1 and σ3 in MRV, the interactions between them and the formation of heterohexmeric complexes are critical for the assembly and disassembly of the reovirus. In this study, 12 VP7 cleavage fragments with 66% coverage were first obtained using in-gel tryptic digestion in the MS assay. To capture the early protease cleavage status, limited tryptic digestion of GCRV was conducted via specialized in vitro trypsin treatment conditions. SDS-PAGE analysis (Figure 6B) clearly showed a strong band of approximately 20 kDa in 15m-TTP and 30m-TTP, while the ~25 kDa and ~18 kDa bands were markedly weaker in 30m-TTP compared with 15m-TTP. This indicated that early cleavage of GCRV occurred close to the C-terminus of VP7. We were unable to clearly identify the early cleaved VP7 fragment position in the N-terminal or C-terminal regions using IB in the current research due to the lack of VP7N and VP7C antibodies. However, based on the resolved GCRV 3D structure, C-terminal VP7 is cleaved first because residues 3 to 88 were resolved in native degraded GCRV particles [25]. Moreover, the remaining VP7 molecules at the N-terminus in 30m-TTP of GCRV and their interaction sites with VP5 were first observed in our recent cryo-EM and 3D image at near-atomic resolution. This showed that the removal of VP7 is not a synchronous event at 5-fold and 2-to-3-fold axes (Wang et al., unpublished data). Moreover, the resolved interaction site between VP5 and VP7 is similar to the MRV μ1 and σ3 interface, which was resolved using the crystal structure [2]. In addition, VP5C/ δ could be detected in 15m-TTP, 30m-TTP, and the two top component samples, and no cleavage fragment was detected with NP, indicating that stepwise cleavage of VP7 could cause VP5 conformational change. A recent study in the temperature-sensitive (ts) MRV mutant tsG453 containing mutations in the S4 gene encoding σ3 indicated that a single point mutation, namely, Asn16 to Lys, is responsible for the ts phenotype of tsG453 [51], which produces only core-like particles. The inability of this mutant to assemble intact virions may be attributed to the disrupted N-terminal interaction of mutant σ3 with μ1, because Asn16 of σ3 is located at the interface with μ1 [51]. Collectively, our results and the well-documented MRV μ1 and σ3 functions suggest that the removal of VP7/σ3 and interactions between μ1/VP5 and σ3/VP7 are significant for initiating a productive infection and causing morphogenesis in infected host cells. 

## 5. Conclusions

In summary, the molecular characteristics of GCRV VP5 and VP7 were investigated in the present study. LC-MS/MS assays and extended biochemical data indicated that cleaved VP5N and VP5C at the autocleavage site Asn42-Pro43 were major components of the particles, which suggested that VP5N and VP5C represent the functional conformations of VP5. In addition, the presence of Thr165, Thr171, Ser283, and Ser556 phosphorylation and two lysine acetylations (Kac219 and Kac623) in VP5 suggested that PTMs may play additional biological roles in VP5 function during particle assembly and infection. Furthermore, stepwise cleavage of VP7 and enhanced infectivity were detected via limited early tryptic treatment, suggesting that multiple VP7 cleavages are essential for VP5 conformational rearrangement and membrane penetration. To the best of our knowledge, the presence of serine and threonine phosphorylation, as well as lysine acetylation in GCRV VP5, are the first evidence of PTMs in reovirus structural proteins. Our results provide significant information for further functional studies on the roles of VP5 PTMs and their association with VP7 in the reovirus life cycle.

## Figures and Tables

**Figure 1 viruses-14-01032-f001:**
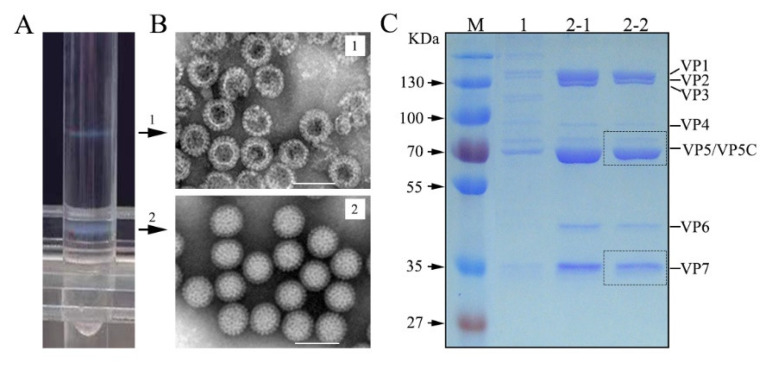
Purified particles and structural protein identification of GCRV. (**A**) CsCl gradient (ρ, 1.20–1.45 g/cm^3^) centrifugation to separate GCRV particle components using a SW41 Ti tube (~12 mL). Arrowheads indicate the position corresponding to the upper and lower band in each gradient. (**B**) TEM of negatively stained virions. 1, top component particles/double empty capsid shell; 2, intact virion. (**C**) SDS-PAGE analysis of structural proteins of purified GCRVs. M, standard protein marker; lane 1 (5 µL), protein components in the upper band; lanes 2-1 (6 μL) and 2-2 (4 μL), protein components in the lower band. The scale bars represent 100 nm. The VP5 and VP7 bands in the dashed box were prepared for in-gel digestion and LC-MS/MS assays. GCRV, grass carp reovirus; LC-MS/MS, liquid chromatography-tandem mass spectrometry; TEM, transmission electron microscopy; SDS-PAGE, sodium dodecyl sulfate-polyacrylamide gel electrophoresis.

**Figure 2 viruses-14-01032-f002:**
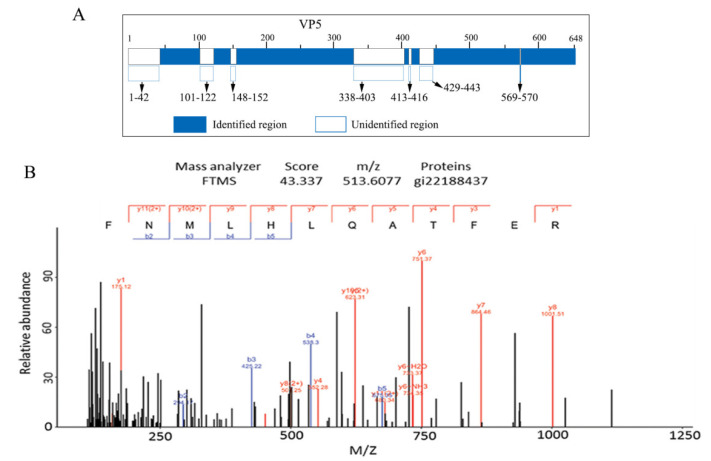
Identification of the tryptic peptide of GCRV VP5 using LC-MS/MS. (**A**) Diagrammatic representation of VP5 tryptic fragments accounted for 492 of the 648 predicted aa in VP5 (76% coverage). (**B**) LC-MS/MS spectra of a tryptic peptide ion of VP5 fragment using in-gel tryptic digestion. GCRV, grass carp reovirus; LC-MS/MS, liquid chromatography-tandem mass spectrometry.

**Figure 3 viruses-14-01032-f003:**
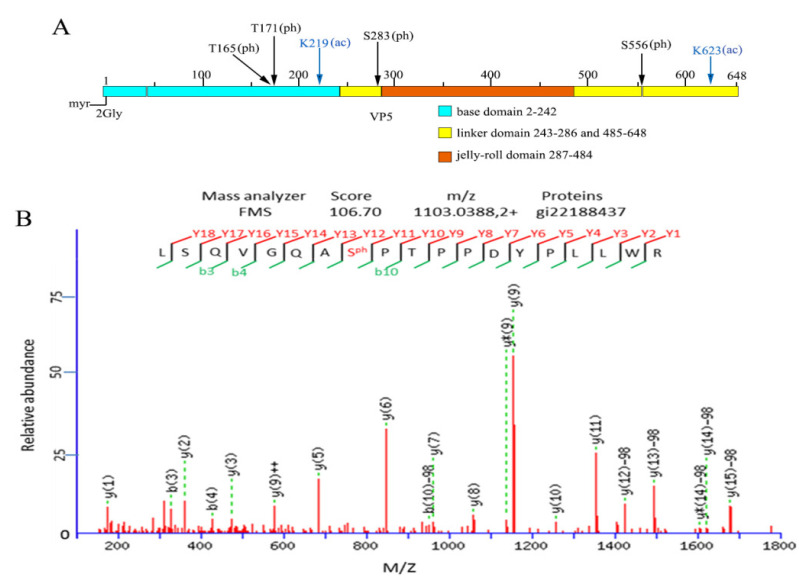
Analysis of phosphorylation and acetylation in VP5 tryptic peptide. (**A**) Diagrammatic representation of identified protein posttranslational modification sites in three different VP5 domains. (**B**) LC-MS/MS spectrum of a tryptic peptide ion Ser556 phosphorylated peptide LSQVGQAS556(ph)PTPPDYPLLWR. LC-MS/MS, liquid chromatography-tandem mass spectrometry.

**Figure 4 viruses-14-01032-f004:**
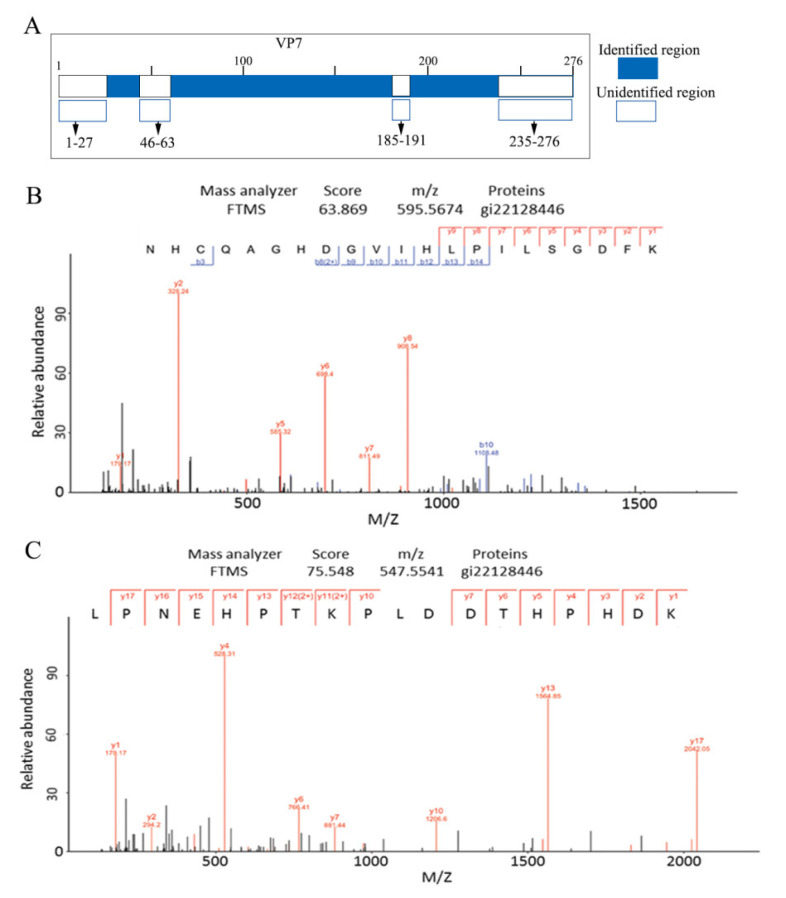
Identification of the GCRV VP7 tryptic peptide using LC-MS/MS. (**A**) Diagrammatic representation of tryptic fragments of VP7. The fragments accounted for 182 of the 276 predicted amino acids in VP7 (66% coverage). (**B**,**C**) LC-MS/MS spectra of two cleaved VP7 peptide fragments using tryptic in-gel digestion. GCRV, grass carp reovirus; LC-MS/MS, liquid chromatography-tandem mass spectrometry.

**Figure 5 viruses-14-01032-f005:**
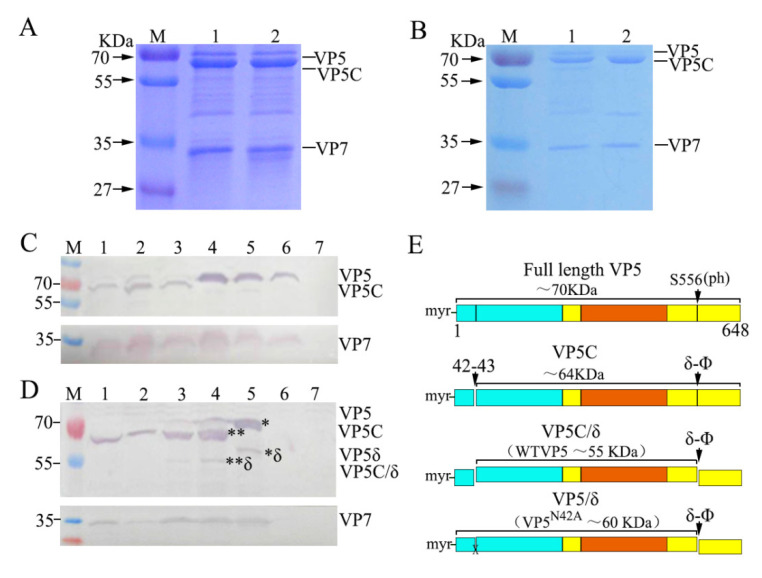
Identification of VP5 and VP5C conformations in GCRV particles and infected cells. (**A,B**) VP5 conformations were analyzed using SDS-PAGE, followed by Coomassie staining from purified virions in sucrose (**A**) and CsCl (**B**) gradient centrifugation. (**C**) Different VP5 conformations were detected using IB in purified VP5/VP7 and VP5^N42A^/VP7 preparations. 1–3, WT-VP5/VP7 (vAcGCRV-VP5/VP7); 4–6, VP5^N42A^/VP7 (vAcGCRV-VP5^N42A^/VP7); 7, mock−infected Sf9 cells. (**D**) Different VP5 conformations were detected using IB in GCRV-infected CIK cells and Sf9 cells co-expressing WT-VP5/VP7 and VP5^N42A^/VP7. 1 and 3, VP5/VP7 in infected CIK cells; 2 and 4, VP5/VP7 in infected Sf9 cells; 5, mutant VP5^N42A^/VP7 in infected Sf9 cells; 6–7, mock-infected cells. ** in D indicates a VP5C/ δ fragment, and * in D indicates a VP5/ δ fragment. (**E**) Diagram of native/WT-GCRV VP5 and VP5^N42A^ protein conformations and their cleavage products identified in (**A**–**D**). Blue, base domain (2–242); yellow, linker domain (243–286 and 485–648); red, jelly−roll domain (287–484). GCRV, grass carp reovirus; SDS-PAGE, sodium dodecyl sulfate-polyacrylamide gel electrophoresis; WT, wild type; CIK, *Ctenopharyngodon idellus* kidney; Sf9, *Spodoptera frugiperda*.

**Figure 6 viruses-14-01032-f006:**
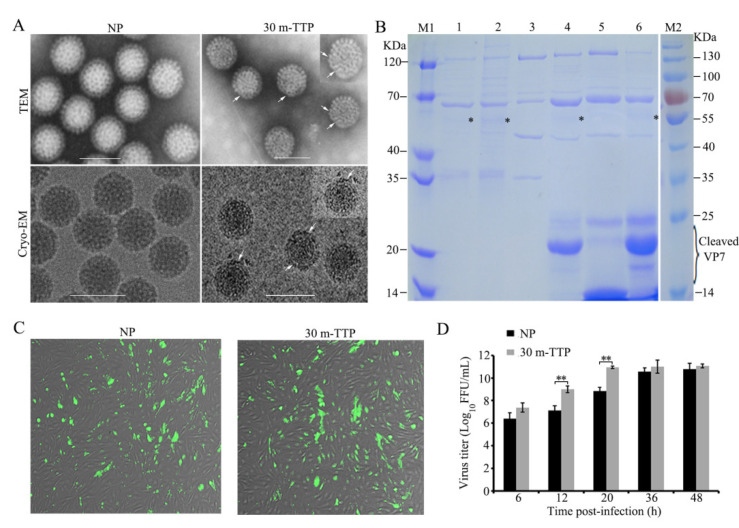
Analyses of early cleavage particle component of GCRV with limited tryptic digestion. (**A**) TEM and cryo-EM of NP and 30m-TTP of GCRV. Arrowheads indicate degraded outer shell protein subunits of 30m-TTP. The 30m-TTP was generated from purified NP preparations of GCRV (~1.5 μg/mL) by treating with trypsin (final concentration: ~100 mg/mL) at 28 °C for 30 min, and then the treated and un-treated preparations were directly applied to a quantifoil grid for TEM, cryo-EM, and inoculation on CIK cells for infectious assays without conducting further purification. (**B**) GCRV particle with limited tryptic-digestion was resolved using SDS-PAGE. M1 and M2, different standard protein marker; 1–2, top component/empty particle; 3, NP of GCRV; 4, 15m-TTP; 5, 30m-ChTP; 6, 30m-TTP; * indicates a VP5C/ δ fragment. (**C**,**D**) Infectivity assays with 30m-TTP and NP of GCRV: CIK cells were infected with 30m-TTP and NP of GCRV, respectively. At 6 h p.i., cells were fixed and stained with anti-VP5 antibody and Alexa-488-labeled donkey anti-mouse IgG to visualize viral replication. Representative images of a fluorescent focus unit assay (**C**). Viral titers were determined at different infection time points (**D**). Data shown are the mean ± standard deviation of three individual experiments. Statistical significance is indicated by asterisks (**, *p* < 0.01). GCRV, grass carp reovirus; TEM, transmission electron microscopy; cryo-EM, cryo-electron microscopy; SDS-PAGE, sodium dodecyl sulfate-polyacrylamide gel electrophoresis; TTP, trypsin-treated particle; 30m-TTP, 30 min trypsin treated particle; NP, native particle; ChTP, chymotrypsin-treated particle.

**Table 1 viruses-14-01032-t001:** Identification of Peptide Fragments with Phosphorylated and Acetylated Modifications in GCRV-VP5.

Fragment#	Start-End	Peptide Sequence	Type of Modification/Site	Mass Charge Ratio (*m*/*z*)
1 1	162–184 162–184	HLDTAMTMLTPDISAGSASCfigNWK HLDTAMTMLTPDISAGSASCNWK	Phosphorylation/Thr_165_ Phosphorylation/Thr_171_	2181.5347, 2+ 849.3580, 3+
2	215–226	YPALKPGNPDTK	Acetylation/Lys_219_	671.8549, 2+
3	270–292	DLDLIEADTPLPVSVFTPSLAPR	Phosphorylation/ser_283_	849.4308, 3+
4	549–567	LSQVGQASPTPPDYPLLWR	Phosphorylation/Ser_556_	1103.0388, 2+
5	616–625	GVTDASEKLR	Acetylation/Lys_623_	559.2952, 2+

## Data Availability

All data are presented in the paper.

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
