# Peer review of "Molecular Characterization of Outer Capsid Proteins VP5 and VP7 of Grass Carp Reovirus"

_viruses, 2022, doi:10.3390/v14051032_

Round 1

Reviewer 1 Report

The authors investigate the characteristics of the grass carp reovirus (GCRV) outer capsid proteins VP5 and VP7 using in-gel tryptic digestion and mass spectrometry (MS). MS analysis identified peptides representing 76% and 66% of the VP5 and VP7 sequences, respectively, and found four phosphorylation sites and two lysine acetylation sites in VP5. Biochemical analysis of native GCRV particles, as well as VP5/VP7 expressed via baculovirus identified cleavage products suggesting that VP5 is cleaved to VP5N/VP5C as the major component of particles. Trypsin digestion of particles resulting in cleavage of VP7 enhanced GCRV replication.

Overall, the manuscript is straightforward and well-presented. The data is generally clear and compelling, though of modest significance. Some minor inconsistencies could be addressed, as outlined below.

  • 1C: What is the difference between lanes 2 and 3? If lane 1 is “top component,” shouldn’t the major structural bands be present, or was much less sample (based on particle number) loaded? (The particle numbers loaded in each well could be included for comparison).
  • 5A/D: What is the difference between lanes 1 and 2 (5A), and between 1/3 and 2/4 (5D)? Stronger VP5Cd bands are observed in lanes 3 and 4, so is there a difference in how the samples were processed?
  • Line 449-450, Fig. 6C: The authors state the level of fluorescent foci in cells following infection by 30m-TTP particles “was obviously enhanced compared to NP.” This does not appear to be the case based on my viewing of 6C. Quantification of % of cells infected would help to solidify this claim.
  • 6D: In a longer time-course of infection, you would expect titers of NP-infected cells to reach equivalence with 30m-TTP particles, suggesting that it is just an effect of enhanced infectivity of the initial inoculum, and not a difference in replicative capacity. It would be important to show these later times 48/72h post-infection to solidify this claim.
  • 6B: The meaning of the asterisk should be denoted in the figure legend.

Author Response

Dear editor and reviewers;

We would like to thank you for critical evaluation of our manuscript, and also for the constructive suggestions/comments.

As you may see in the revised version of our manuscript, we have addressed all of your suggestions and comments, and revised our manuscript accordingly. For the cell lines related information, we have made the supplement in this revised manuscript. Responses/answers to reviewer’s comments are also enclosed within this letter. All text modifications are marked up using the “Track Changes” function and highlighted by red colour in the revised manuscript. For your convenience, the questions/suggestions are shown in black and our responses are in blue.

Thanks a lot for your time and consideration. We are looking forward to hearing from you soon.

Sincerely yours,

Qin Fang

Reviewer 2 Report

In the MS, the authors investigated molecular characteristics of VP5 and VP7, the outer capsid proteins of GCRV by LC-MS/MS, and explored the biological functions of VP5 and VP7 by studying the related biochemical properties of VP5 and VP7. The MS can be accepted after the following questions will be addressed.

  1. In the biological classification, the Latin writing of “family”and above is written in normal, not italic.
  2. Line 134-136, Why to use a spectrophotometer to measure the concentration of purified virus.What is the principle? The authors need provide the reference(s).
  3. There are someproblems in English writing in this MS, and the authors need avoid them. For example, Lines 142, there is a slip of the pen in “the S6 and S10 GCRV genes of were...” . Lines 293-296, the sentence needed to be written; Line 300, the authors also use two “-ing” forms in a row and it is better to change another word.
  4. Line 287-289, Why is just one fragment not cleaved at the specific tryptic cleavage site? The authors need provide some explanations.
  5. Line 293, the expression "two to three" is not very accurate and it should be clearly stated that how many are there.
  6. Line 315, there are many kinds of posttranslational modifications of proteins. Why did the authors choose phosphorylated and acetylated modifications to analyze?
  7. Line 451-452, the authors express that “the infectivity increased more than 100-fold with 30m-TTPs infection compared to mock-treated NPs”, while the significance of the difference is not marked in the Figure 6D.

Author Response

(The authors gave the same response as above.)
